# Total Phenolic Content, Biomass Composition, and Antioxidant Activity of Selected Marine Microalgal Species with Potential as Aquaculture Feed

**DOI:** 10.3390/antiox11071320

**Published:** 2022-07-04

**Authors:** Vasilis Andriopoulos, Maria D. Gkioni, Eleni Koutra, Savvas G. Mastropetros, Fotini N. Lamari, Sophia Hatziantoniou, Michael Kornaros

**Affiliations:** 1Department of Chemical Engineering, School of Engineering, University of Patras, 26504 Patras, Greece; billandri@upatras.gr (V.A.); ekoutra@chemeng.upatras.gr (E.K.); savvasgiannismas@gmail.com (S.G.M.); 2Department of Pharmacy, School of Health Sciences, University of Patras, 26504 Patras, Greece; marigion00@gmail.com (M.D.G.); flam@upatras.gr (F.N.L.); sohatzi@upatras.gr (S.H.)

**Keywords:** antioxidant activity, DHA, DPPH radical-scavenging activity, EPA, FRAP, GC-MS, in situ transesterification, iron chelating activity, Pearson correlation, principal component analysis

## Abstract

There has been growing interest in microalgal biomolecules for health and cosmetics, as well as in the use of microalgae as aquaculture feed due to the need to replace fishmeal and fish oil with sustainable yet equally nutritious alternatives. Aim of this study is to evaluate the potential of five marine microalgal species, namely *Chlorella minutissima*, *Dunaliella salina*, *Isochrysis galbana*, *Nannochloropsis oculata* and *Tisochrysis lutea*, for the co-production of antioxidants and aquaculture feed. Batch cultivation was performed under saturating light intensity and continuous aeration. Freeze-dried biomass was extracted sequentially with water and methanol and evaluated for phenolic content and antioxidant activity, as well as proximate composition and fatty acid profile. Methanolic extracts of *C. minutissima* presented the highest phenolic content, measured with the Folin–Ciocalteu assay, and antioxidant activity. However, HPLC and LC-MS showed the presence of non-pigment compounds only in *T. lutea*. Total phenolic content and antioxidant activity were correlated to chlorophyll content. *N. oculata* and *T. lutea* were rich in eicosapentaenoic acid and docosahexaenoic acid, respectively, as well as in protein. In conclusion, *N. oculata* and *T. lutea* are suitable candidates for further optimization, while the data presented suggest that pigment effects on the Folin–Ciocalteu method require reconsideration.

## 1. Introduction

Microalgal secondary metabolites have been the focus of extensive scientific research, while a few of them, mainly belonging to carotenoids [1], are currently produced at a large scale successfully. Microalgal carotenoids such as astaxanthin and fucoxanthin, which are important for the pigmentation and health of farmed fish [2,3], can also modulate important metabolic pathways such as NAAA-PEA and Nrf2-ARE [4,5]. The lipid fraction of many microalgal species is rich in unsaturated fatty acids, which apart from very important nutritional value can have a variety of anti-inflammatory and antioxidant activities [6]. Apart from pigments and lipids, other microalgal biomolecules such as phenolics, flavonoids, sterols, and tocopherols have been the subject of many recent studies due to their antioxidant, anti-inflammatory, and antimicrobial properties [7,8,9]. More specifically, phenolics represent a wide group of chemical compounds, mainly derived from plants, which offer significant advantages upon consumption, including prevention of several health disorders due to their radical scavenging activity [10]. Interestingly, various phenolic and flavonoid compounds such as chlorogenic acids, coumarines, flavanols, flavanones, flavones, hydroxybenzoic acids, hydrocinnamic acids, and their derivatives (esters, glycosides etc.,) have been identified in several microalgal species such as *Arthrospira maxima*, *Euglena cantabrica*, *Chlorella* sp., *Phormidium* sp., *Tetraselmis* sp., *Isochrysis* sp., *Phaeodactylum* sp., and others [11,12]. Klejdus et al. identified benzoic and cinnamic acid derivatives in *Spongiochloris spongiosa* and several cyanobacterial species [13]. Scaglioni et al. quantified small phenolic acids in methanolic and ethanolic extracts of *Spirulina* sp. and *Nannochloropsis* sp. and correlated their concentration to antioxidant activity against 2,2′-azino-bis(3-ethylbenzothiazoline-6-sulfonic acid) (ABTS) and 2,2-diphenyl-1-picrylhydrazyl (DPPH) [14]. Sozmen et al. quantified 30 phenolic and flavonoid compounds in ethanolic extracts of *Chlorella miniata* cultivated under different temperatures and light intensities and reported that ellagic acid and quercetin correlated to antioxidant activity against DPPH and Ferric Reducing Antioxidant Power Assay (FRAP) activity [15]. Strejckova et al. measured changes in abundance of rosmarinic acid and other phenolic compounds in *Scenedesmus quadricauda* in response to heavy metal concentration [16]. UV radiation and nitrogen limitation have also been reported to induce accumulation of phenolic compounds, measured with the Folin–Ciocalteu assay, in *Dunaliella salina* and *Arthrospira platensis* [17]. Mishra et al. observed a decrease of total phenolic content (TPC), measured with the Folin–Ciocalteu assay, and DPPH radical scavenging activity (RSA) in response to high light intensity and nitrogen limitation in methanolic extracts of *I. galbana* [18]. It has to be noted that the phenolic content of microalgae quantified with chromatographic methods is in the order of μg g^−1^ dry weight (DW), while the total phenolic content registered by the Folin–Ciocalteu assay is in the order of mg g^−1^ DW [8,19]. Additionally, there is an ambiguity in the contribution of phenolic compounds on the antioxidant activity of microalgae since carotenoids also have a major effect [20]. Since there is yet no consensus on the magnitude of the phenolic content of microalgae, as well as their contribution to the antioxidant and reducing capacities of the derived extracts, further research is needed until microalgae-derived phenolics become a viable target product.

Furthermore, one major application of microalgae is aquaculture, which represents a growing industry that paradoxically puts a burden on wild fish reservoirs since the protein and lipid nutritional demands of farmed fish are primarily satisfied by fish meal and fish oil [21]. Currently, fish meal and fish oil have been partially replaced by plant-derived substitutes which however lack eicosapentaenoic acid (EPA) and docosahexaenoic acid (DHA), the two most important polyunsaturated fatty acids (PUFAs) for human nutrition that are obtained primarily by fish, as well as some essential amino acids [21]. Microalgae have been proposed as a more suitable alternative since they contain high quality protein and are the primary EPA and DHA producers in nature, while live microalgae are already widely used in hatcheries to feed fish larvae or crustaceans and rotifers that are then fed to fish [22]. However, large-scale microalgal applications require a multi-product approach in order to be economically feasible [23], for example with a high-value product, such as antioxidants, being produced along a bulk product, such as aquaculture feed.

Therefore, the aim of the present study was to provide more information on the phenolic content of selected marine microalgal species that, based on literature, have a promising potential for phenolics production, as well as for the utilization as aquaculture feed. Furthermore, an emphasis is given on biomass composition and fatty acid analysis, along with antioxidant activity testing with view to unraveling the potential uses of promising marine microalgae.

## 2. Materials and Methods

### 2.1. Chemicals and Reagents

Macronutrients used were NaNO_3_ (PanReac, Barcelona, Spain) and NaH_2_PO_4_·2H_2_O (Honeywell International Inc., Charlotte, NC, USA), while the trace elements used for media preparation were Na_2_EDTA (Sigma-Aldrich, St. Louis, MO, USA), FeCl_3_·6H_2_O (Acros Organics, Geel, Belgium), CuSO_4_·5H_2_O (Sigma-Aldrich), ZnSO_4_·7H_2_O (Sigma-Aldrich), CoCl_2_·6H_2_O (Thermo Fisher Scientific, Pittsburg, PA, USA), MnCl_2_·4H_2_O (Acros Organics), Na_2_MoO_4_·2H_2_O (Chem-Lab NV, Zedelgem, Belgium). Cyanocobalamin, Thiamine HCl and Biotin were purchased from Sigma-Aldrich.

Chemicals used for analysis were ammonium bicarbonate (Sigma-Aldrich), Folin–Ciocalteu reagent (Sigma-Aldrich), N,N′-dimethylformamide (DMF) (Honeywell), chloroform HPLC grade (Honeywell), mercury (III) oxide red (Sigma-Aldrich), 2,4,6-tri(2–pyridyl)–s–triazine (TPTZ) (Alfa Aesar, Ward Hill, MA, USA), water for analysis (Carlo Erba, Cornaredo, Milan, Italy), methanol HPLC grade (Thermo Fischer Scientific), sodium acetate trihydrate (Merck), acetic acid (PENTA), hydrochloric acid (HCl) (PENTA), 1,1-diphenyl-2-picrylhydrazyl radical (DPPH∙) (Sigma-Aldrich), iron (II) chloride anhydrous (Acros organics), 3-(2-pyridyl)-5,6-diphenyl-1,2,4-triazine-p,p′-disulfonic acid monosodium salt hydrate (ferrozine) (Sigma-Aldrich), EDTA-Na_2_, 6-hydroxy-2,5,7,8-tetramethylchroman-2-carboxylic acid (Trolox) (Acros organics).

### 2.2. Microalgal Species and Cultivation Conditions

*Chlorella minutissima* (Foti & Novak), *Isochrysis galbana*, *Nannochloropsis oculata* and *Tisochrysis lutea* were provided by the Laboratory of Zoology (Department of Biology, University of Patras, Greece), while *Dunaliella salina* (184.80) was obtained from the SAG Culture Collection (University of Göttingen). All species were cultivated in four times concentrated f/2 medium without silicate, at an ambient temperature of 25 °C, under continuous illumination of ~350 μmol m^−2^ s^−1^ provided by 6000 K white LED light bulbs placed below the cultures. Erlenmeyer flasks of 500 mL capacity and operational volume of 400 mL were inoculated with 0.1–0.16 g L^−1^ biomass, grown under the same conditions as mentioned above, until the end of the exponential growth phase. Initial pH was adjusted to 8 with the addition of 1 Ν NaOH before the inoculation. Ambient air was provided at a rate of ~2.8 L L^−1^ min^−1^ which was also the sole means of mixing. Samples were obtained every 2–3 days to monitor pH, as well as biomass concentration via optical density at 750 nm (OD_750_) and total suspended solids (TSS) determination. Nitrate and phosphate concentrations of the culture medium were determined at the end of each run.

### 2.3. Analytical Measurements

TSS were measured according to *Standard Methods for the Examination of Water and Wastewater* [24], by using 0.5 M ammonium bicarbonate for biomass washing [25]. OD_750_ was measured with a Cary50 UV/VIS, Varian spectrophotometer, which was also used to derive the maximum specific growth rate (μ_max_) during the logarithmic growth phase. Standard curves of TSS vs. OD_750_ [26] were produced for every species at different times during the duration of the experiment and used for the determination of the biomass concentration (Dry weight basis or DW). Nitrates in the cultivation medium, obtained after cells removal (via filtering through Whatman, GF/F filters) at the end of each run, were measured spectroscopically at 220 and 275 nm [24], while total phosphorus was measured as orthophosphates with the ascorbic acid method after hydrolyzation under low pH [24].

### 2.4. Biomass Composition Analysis

#### 2.4.1. Biomass Collection

At the end of each growth period, wet biomass was obtained via centrifugation at 3780× *g* for 7 min (Z 366, Hermle AG, Gosheim, Germany), washed with 0.5 M ammonium bicarbonate [25] and freeze dried (Telstar, LyoQuest, Barcelona, Spain).

#### 2.4.2. Moisture and Ash Content Determination

Moisture content of the freeze-dried biomass was determined after drying at 105 °C [24], and ash content was determined after further incineration at 550 °C for 45 min [24].

#### 2.4.3. Protein Content Determination

Protein content was determined with the Semi-micro Kjeldahl method [24]. Specifically, 20 mg of freeze-dried biomass were digested with 7 g K_2_SO_4_, 350 mg H_g_O, 50 mL 3D H_2_O, and 10 mL concentrated H_2_SO_4_ at 200 °C for 1 h and at 370 °C for 2 h using a VELP Scientifica Automatic Digestion Unit. Produced ammonium was then converted to free ammonia and captured in 25 mL of 2% boric acid with pH indicators using a VELP Scientifica Kjeldahl Distillation unit. The captured ammonia was quantified with titration using 0.02 N H_2_SO_4_ and converted to protein content with a conversion factor 6.25 [24].

#### 2.4.4. Carbohydrate Content Determination

Carbohydrate content was determined with the phenol-sulfuric acid method [27]. Specifically, 5–10 mg of freeze-dried biomass were suspended in 10–100 mL 3D H_2_O. While stirring with a magnet, 1 mL of the suspension was transferred to a glass vial and digested with 1 mL 5% phenol and 5 mL concentrated H_2_SO_4_. Absorption was measured at 490 nm and converted to glucose equivalents with a standard curve [27].

#### 2.4.5. Lipid Profiling and Quantification

Lipid content and the lipid profile were determined with one-step in situ transesterification [28] and subsequent analysis of the derived fatty acid methyl esters (FAMEs) was carried out on a GC (7890A, Agilent Technologies Inc., Santa Clara, CA, USA), equipped with a flame ionization detector (FID) and a capillary column (DB–WAX, 10 m × 0.1 mm × 0.1 μm) as previously described [29]. A mixture of 22 fatty acids (C14:C22) was used as the standard, with C17:0 as the reference used for correction [29].

#### 2.4.6. Pigment Content Determination

Chlorophyll a and b (Chla, Chlb) as well as total carotenoids were determined via extraction with N,N′-dimethylformamide (DMF), at room temperature, for at least 20 min and subsequent spectroscopic estimation according to previous studies [30,31].

### 2.5. Extraction Methods

#### 2.5.1. Sequential Extraction with H_2_O and MeOH (Protocol A)

This extraction method was used to separately measure the contribution of water-soluble molecules and pigments to the antioxidant activity of the microalgal species under study. The extraction took place as follows: ~50 mg of freeze-dried biomass were vigorously mixed with 2 mL of 3D H_2_O and incubated for 20 min at 80 °C under frequent mixing. The supernatant was obtained after centrifugation at 3780× *g* for 5 min, filtered with a 0.2 μm syringe filter (Whatman^®^ Puradisc 25), and collected in a glass vial. The process was repeated two more times. Subsequently, the process was ×3 repeated using room temperature methanol instead of hot 3D H_2_O as the solvent. Aqueous and methanolic extracts were collected separately and stored at −18 °C under nitrogen atmosphere.

#### 2.5.2. Ultrasound-Assisted Extraction with 70% MeOH (Protocol B)

Approximately 20 mg of freeze-dried biomass were extracted with 2 mL 70% MeOH for 15 min in an ultrasonic bath at a temperature maintained below 40 °C. The extraction was repeated two more times and extracts were collected in the same vial. Non-polar compounds were removed with liquid–liquid extraction with cyclohexane. Extracts were condensed with a rotary-evaporator, freeze-dried, and stored at −18 °C until chromatographic analysis.

#### 2.5.3. Ultrasound-Assisted Extraction with MeOH (Protocol C)

This extraction method was used to increase the concentration of extracted polar compounds. In summary, 200 mg of freeze-dried biomass were extracted with pure MeOH in the same biomass to solvent ratio as Protocol B for 15 min in an ultrasonic bath with the temperature maintained below 40 °C. The extraction was repeated two more times and extracts were collected in the same vial. Non-polar compounds were removed with liquid–liquid extraction (cyclohexane). Extracts were condensed with a rotary-evaporator, freeze-dried, and stored at −18 °C until chromatographic analysis.

### 2.6. Determination of Total Phenolic Content (TPC)

TPC of the extracts prepared with Protocol A was determined with a modified Folin–Ciocalteu method [32]. More specifically, 100 μL of extract were added to 3D H_2_O to a final volume of 7 mL. Subsequently, 0.5 mL of Folin–Ciocalteu reagent was added. After 1–2 min, 1.5 mL 1.89 M Na_2_CO_3_ solution was added. The volume was finally adjusted to 10 mL with 3D H_2_O. After incubation of 2 h in the dark, absorption at 760 nm was measured and converted to mg gallic acid equivalents (GAE) L^−1^ with a standard curve. The results were also expressed as mg GAE per g of biomass DW (mg GAE g^−1^ DW).

### 2.7. Determination of Antioxidant Activity

#### 2.7.1. Ferric Reducing Antioxidant Power (FRAP) Assay

The FRAP assay was used to measure the ability of antioxidants to reduce the [Fe(TPTZ)2]^3+^ to [Fe(TPTZ)2]^2+^ [33]. Specifically, 10 μL sample or standard solution, in this case Trolox, was placed in a 96-well microplate followed by the addition of 190 μL FRAP reagent. FRAP reagent was freshly prepared with 300 mM sodium acetate/acetic acid buffer pH 3.6, 10 mM TPTZ in 40 mM HCl, and 20 mM ferrous chloride hexahydrate at a ratio of 10:1:1 respectively. The plate was incubated at 37 °C for 5 min and afterwards the absorbance was measured at 595 nm using a TECAN sunrise microplate reader. The results were expressed as mg Trolox equivalent (TE) per g of biomass DW (mg TE g^−1^ DW), based on the plotted calibration curve of the standard Trolox.

#### 2.7.2. DPPH Radical Scavenging Activity (DPPH RSA) Assay

The assay which has been chosen for the determination of the radical scavenging activity of the samples was the scavenging of the DPPH radical [34]. Briefly, 20 μL sample or standard solution were placed in a 96-well microplate. About 80 μL of 4 mM DPPH methanolic solution was added and the plate was kept in the dark and ambient temperature, for 30 min. The absorbance was measured at 540 nm. Trolox was selected as the standard in that assay. The percentage of the radical scavenging activity was calculated by the equation:%RSA = (A_blank_ − (A_sample_ − A_control_)/A_blank_) × 100

The results were expressed as mg Trolox equivalent per g of DW (mg TE g^−1^ DW), based on the linear area of the plotted calibration curve of Trolox.

#### 2.7.3. Iron Chelating Activity (ICA) Assay

ICA determination was based on a previous method with slight modifications [35]. A total of 100 μL sample was placed in a 96-well microplate, followed by the addition of 50 μL of 2 mM iron (II) chloride. The addition of 20 μL ferrozine solution 5 mM resulted in the onset of the reaction. The formation of ferrozine-Fe^2+^ complex leads to a magenta color. The change in color was measured after 30 min at 540 nm. EDTA-Na_2_ was used as the positive control. The results were expressed as % iron chelating capacity (ICA) and calculated by a relevant equation with DPPH assay:%ICA = (A_blank_ − (A_sample_ − A_control_)/A_blank_) × 100

They were also expressed as mg EDTA g^−1^ DW.

### 2.8. Chromatographic Methods

#### 2.8.1. High Performance Liquid Chromatography (HPLC)

An Agilent HPLC (Series 1260, Agilent Technologies Inc., Santa Clara, CA, USA) equipped with DAD detector and a POROSHELL 120 EC-C18 (4.6 mm × 150 mm, 2.7 μm) column was used in dual-wavelength mode, 260 nm for monitoring of aromatic compounds and 405 nm for monitoring of pigments in hydromethanolic and methanolic extracts (Protocols B & C). Sample volume was 20 μL and flow rate was maintained at 1 mL min^−1^. Elution was gradual, with 0.2% formic acid in 3D H_2_O (A) and 0.2% formic acid in acetonitrile (B). Initially, the elution was isocratic for 5 min with 5% solution B, followed by a gradual increase to 43% B in 10 min which was maintained for 5 min and then gradually increased to 50% B in 5 min. Finally, the percentage of B in the elution solution gradually increased to 100% in 7 min, ran isocratically for 5 min, and gradually decreased to 5% in 2 min and ran isocratically until the end. Temperature of the column was maintained at 30 °C.

#### 2.8.2. Liquid Chromatography-Mass Spectrometry (LC-MS)

An Agilent LC 1260 Infinity II equipped with Single Quadrupole mass analyzer (Agilent Technologies) and a POROSHELL 120 EC-C18 (4.6 mm × 100 mm, 2.7 μm) column was used. Sample volume was 10 μL and flow rate was maintained at 0.5 mL min^−1^. The rest of the procedure was the same as the HPLC protocol above. For the analysis of phenolic standards and *T. lutea,* a different elution program was used: 0% B for 5 min, gradual increase to 20% in 2 min, gradual increase to 41% in 10 min, gradual increase to 43% in 15 min, gradual increase to 80% in 15 min which was maintained for 5 min, gradual increase to 95% in 10 min which was maintained for 5 min, and finally gradual decrease to 0% in 10 min.

#### 2.8.3. Semi-Preparative HPLC

An Agilent HPLC (Series 1260, Agilent Technologies Inc., Santa Clara, CA, USA) equipped with DAD detector and a Phenomenex Luna^®^ C18 (10 mm × 250 mm, 5 μm) column was used in dual-wavelength mode (260 and 405 nm). Sample volume was 100 μL and flow rate was maintained at 2 mL min^−1^. The mobile phase was composed of: (A) 3D H_2_O with 0.1% formic acid and (B) acetonitrile. The elution was carried out as follows: 7 min 5% B, gradual increase to 100% B in 23 min, 100% B for 10 min, gradual decrease to 5% B in 2 min and 5% B for 8 min.

### 2.9. Statistical Analysis

All the experiments and measurements were performed in duplicate. Average values are presented along with the standard deviation. Analysis of variance (ANOVA) and Principal components analysis (PCA) were performed in Matlab, while Pearson correlation in Microsoft Excel (XLSTA-Cloud). Data were z-score normalized for PCA.

## 3. Results and Discussion

### 3.1. Growth and Biomass Productivity

The highest μ_max_ values were presented by *N. oculata* and *T. lutea* (0.45 ± 0.05 and 0.45 ± 0.11 d^−1^ respectively) (Table 1), without any statistical significance from the rest of the species. The highest final TSS concentration were attained by *C. minutissima* and *N. oculata*, 1.35 ± 0.13 and 1.25 ± 0.05 g L^−1^ DW, respectively, (Table 1, Figure 1), significantly higher than the final TSS of *I. galbana*. OD_750_ was significantly higher for *C. minutissima* and *N. oculata* (6.85 ± 0.18 and 6.21 ± 0.69 respectively) than the rest of the species (Table 1, Figure 1). Manisali et al. have previously reported a μ_max_ between 0.36 and 0.59 d^−1^ for N/P ratio and between 5 and 15 for *N. oculata* [36]. Similarly, the N/P ratio in the current study (9.5), as well as μ_max_ of *N. oculata* fall in those ranges (Table 1). Gao et al. developed an improved strain of *T. lutea* and reported a μ_max_ value of 0.15 and 0.49 d^−1^ for the original and the improved strain, respectively [37]. All species consumed practically all the N in the growth medium, except for *D. salina* which consumed significantly less N than all the species except for *C. minutissima* (Table 1). *D. salina* also consumed significantly less P than *C. minutissima* and *N.oculata* (Table 1). *D. salina* cells were sedimenting much more rapidly than the other species, possibly in response to the elevation of pH [38]. *N. oculata* reached a higher maximum pH than the rest of the species (>10 comparing to <9.4 for the rest of the species), as shown in Figure 2, and the final pH was significantly higher than *T. lutea* (Table 1).

### 3.2. Biomass Composition and FA Analysis

#### 3.2.1. Late Stationary Phase Composition and FA Profile

One of the primary purposes of the present study was the evaluation of the biomass components, including proteins, carbohydrates, lipids, pigments, and inorganic compounds, as well as further analysis of the lipid fraction. As shown in Table 2, the highest ash content was found in *T. lutea* (12.81 ± 3.16%), significantly higher than that of *N. oculata* (3.81 ± 0.21%). No significant differences among species were found for protein and carbohydrate content. *T. lutea* and *N. oculata* presented the highest and lowest protein content, 39.69 ± 5.03% and 27.71 ± 3.12% respectively. The highest carbohydrate content was found in *N. oculata* (21.48 ± 10.62%), while the lowest in *I. galbana* (8.81 ± 2.59%). *C. minutissima* had significantly higher Chl content than the rest of the species, with 2.8 ± 0.16% Chla and 1.01 ± 0.05% Chlb, and higher but not significantly different carotenoid content than *T. lutea*. The carotenoid content of *C. minutissima* (1.07 ± 0.08%) is similar to that reported by Bauer et al. [39], which was attributed primarily to lutein, while that of *T. lutea* (0.77 ± 0.07%) can be attributed to fucoxanthin, which can reach up to 2% of the dry cell weight [37]. *D. salina* contained Chlb (0.29 ± 0.04%) in contrast to *N. oculata* and *T. lutea* [40,41], while *I. galbana* contained traces (0.08 ± 0.03%). In addition, the lowest Chla and carotenoid contents were found in *N. oculata* and *D. salina*, 0.71 ± 0.31% and 0.24 ± 0.08% respectively.

Concerning the intracellular lipids of the selected species, *N. oculata* contained significantly more lipids than the rest of the species (36.7 ± 3.3%), while the lowest lipid content was found in *T. lutea* (7.39 ± 0.45%) (Table 2). Accumulation of lipids upon nitrogen and/or phosphate limitation in *N. oculata* is well established [42]. Regarding the FA profile, as presented in Figure 3, *C. minutissima* and *D. salina* contained predominantly C16:0 and C18 PUFA and no EPA or DHA, a characteristic profile for Chlorophycae [43]. *N. oculata* contained predominantly C16:0 and 16:1 and presented significantly higher EPA content, 28.19 ± 0.03 mg g^−1^ DW than the rest of the species. High fraction of 16C FAs and EPA is characteristic for Eustigmatophyceae such as *Nannochloropsis* [43], with EPA reaching up to 34% of total lipids and 67.6 mg g^−1^ DW [44] in *N. oculata*. *I. galbana* contained predominantly C18:1, DHA (21.56% of total FA and 14.93 ± 2.76 mg g^−1^ DW), C14:0 and C16:0 FAs. *T. lutea* contained the highest DHA content of 21.56 ± 2.04 mg g^−1^ DW, which was not significantly different from that of *I. galbana*, with C14:0 and C16:0 being the other more abundant FAs. Information of total monounsaturated, polyunsaturated, and saturated FA fractions is also provided in Table 2.

#### 3.2.2. Early Stationary Phase Composition and FA Profile of *N. oculata* and *T. lutea*

Since *N. oculata* and *T. lutea* presented both the highest growth rates and EPA and DHA yields, their composition was evaluated at the beginning of the stationary phase as well (Table 3). No significant differences in protein and carbohydrate content were found between the two species during the early stationary phase or between the growth phases for each species. However, protein content at the early stationary phase was 39.88 ± 1.72% and 43.30 ± 1.33% for *N. oculata* and *T. lutea*, ~30 and 9% higher than the content at the late stationary phase respectively. Carbohydrate content at the early stationary phase was 14.92 ± 1.83 for *N. oculata* and 7.21 ± 0.10% for *T. lutea*, ~51.7 and 28.3% lower than the content at the late stationary phase respectively. In addition, Chla content was 1.86 ± 0.11% for *N. oculata* and 1.95 ± 0.11% for *T. lutea* (~2.6 and 2 times higher, respectively), with the difference being significant for *N. oculata.* Carotenoid content followed the same trend, although there were not significantly important differences. No significant differences were found between growth phases for the lipid content and profile. However, the lipid content of *N. oculata* during the early stationary phase was 26.39 ± 3%, ~1.4 times lower than the content at the late stationary phase and significantly higher than the lipid content of *T. lutea*. The EPA content was 33.74 ± 9.98 mg g^−1^ DW, ~1.2 times higher than the content during the late stationary phase. Both the total lipid and DHA content of *T. lutea* (9.45 ± 0.3% and 31.31 ± 2.92 mg g^−1^ DW, respectively) were higher at the early stationary phase, ~1.3 and 1.5 times higher than in the late stationary phase respectively. Decrease in protein, PUFA and Chl content during nitrogen and/or phosphorus starvation is well documented for *N. oculata* [36]. Lastly, Matsui et al. reported that *T. lutea* attains a high DHA content at the decelerating phase which is maintained through the stationary phase [45].

### 3.3. Total Phenolic Content and Antioxidant Activity

The phenolic content of microalgae is usually quantified using the Folin–Ciocalteu (F-C) assay, a non-specific method used to measure the metal reducing capacity of plant extracts [32], which is expressed as mg of gallic acid equivalents per g of sample dry weight, where the sample can be the dried biomass or the dried extract. In the latter case, the extraction yield [46] is seldomly given [47], making comparison of results from different authors difficult. Other antioxidant tests usually accompany the F-C assay, and correlations between the different activities are derived. The antioxidant capacity of extracts is often given in obscure units such as the percentage of scavenging of a given amount of oxidant or IC50, the half maximal inhibitory concentration, of the extract. Apart from the differences in the units of available data, there is also a variation in the results of different authors, with some reporting a correlation of phenolic content with antioxidant capacity [20,47] while others do not [48,49]. Variation in the phenolic content measured by Folin–Ciocalteu in the literature might stem from actual differences in the phenolic content of different species and the effect of different growth conditions; however, it might also be related to the choice of solvents, extraction time [50], and temperature. Regarding the choice of solvent, phenolic compounds are often extracted from the microalgal cells using only methanol. However, methanol extracts pigments, which can make up a significant fraction of the cell (1–14%) [51] and can interfere with the aforementioned tests. In the present study, we chose a two-step extraction method: first using hot water to extract polar compounds like phenolics and subsequently methanol to extract the pigments. TPC, DPPH RSA, FRAP, and ICA of aqueous and methanolic extracts were evaluated for all species at the early and late stationary phase. Results are given in Table 4, and the main findings are summarized below.

#### 3.3.1. TPC

Three-way ANOVA showed significant effects of species, solvent, phase-solvent interaction, and species-solvent interaction on TPC. Aqueous extracts at the late stationary phase presented significantly higher TPC than methanolic extracts at the late stationary phase, while methanolic extracts of *C. minutissima* and aqueous extracts of *D. salina* had significantly higher TPC than the rest of the extracts except for aqueous extracts of *I. galbana*. The highest and lowest TPC in aqueous extract was found in *D. salina* at the late stationary phase (8.78 ± 1.49 mg GAE g^−1^ DW) and *C. minutissima* at the late stationary phase (2.81 ± 0.24 mg GAE g^−1^ DW) respectively. TPC of the aqueous extract of *D. salina* was lower than that reported by Li et al. [48] for *Nostoc ellipsosporum* (~10 mg GA g^−1^ DW) but higher than the other reported values in the literature (Table 5). The methanolic extract of *C. minutissima* at the early stationary phase presented significantly higher TPC than all other methanolic extracts (9.04 ± 0.68 mg GAE g^−1^ DW), while the methanolic extract of *T. lutea* at the late stationary phase (1.25 ± 0.44 mg GAE g^−1^ DW) the lowest. TPC values of 3–24 mg GAE g^−1^ DW have been previously reported in *C. minutissima* [52,53], with comparable results to this study (9–12 mg GAE g^−1^ DW).

The combined TPC (the sum of TPC of aqueous and methanolic extracts) in all the species examined (6–12 mg GAE g^−1^ DW) was in some cases lower [53,54,55,56], similar [8,11,17,48,49,52,56,57], or higher [11,14,15,17,48,49,55,58] than previously reported values. In practical terms, the phenolic content of 5.1–6.85 mg GAE g^−1^ DW presented by *N. oculata* and *T. lutea* was ten times higher than that of herbs like rosemary, sage, and mint (0.5–0.6 mg GAE g^−1^ DW) [59] but lower than that of agri-food waste products like grape vines, tomato waste and grapefruit waste, which have a high phenolic content (32 mg GAE g^−1^ DW for dried grape vines) and strong antioxidant capacity [60]. Since phenolic compounds can be obtained in similar or higher amounts from readily available and inexpensive waste, developing a microalgal process for phenolics production should also target to other products in the context of a biorefinery. Under this scope, phenolics and other bioactive compounds from microalgae can be an excellent co-product of aquaculture feed, as has been recently suggested by techno-economic analysis [61].

**Table 5 antioxidants-11-01320-t005:** Average, maximum and minimum values of TPC for various genera of microalgae and cyanobacteria. The number of references from which the average values were calculated is also provided.

Genus	N of Ref	TPC	Min	Solvent	Ref	Max	Solvent	Ref
*Acutodesmus*	1	6.4		Acetone	[56]			
*Anabaena*	4	6.7 ± 7.0	0.6	Hexane	[49]	24.5	Hexane, Chlorophorm, Ethyl Acetate, Ethanol 70%, Water	[55]
*Arthrospira*	3	6.4 ± 5.4	0.0	Hexane	[14]	22.5	Hexane, Chlorophorm, Ethyl Acetate, Ethanol 70%, Water	[55]
*Botryococcus*	1	10.2 ± 3.1	7.1	Acetone	[56]	13.2	Acetone	[56]
*Chaetoceros*	1	0.4 ± 0.2	0.2	Water	[62]	0.7	Hexane	[62]
*Chlamydomonas*	1	5.0 ± 2.4	2.4	Water	[48]	8.1	Hexane	[48]
*Chlorella*	11	4.8 ± 6.7	0.0	Hexane	[52]	39.1	Hexane, Chlorophorm, Ethyl Acetate, Ethanol 70%, Water	[55]
*Chlorococcum*	1	2.9 ± 0.7	2.2	Water	[47]	3.7	Ethanol	[47]
*Chroococcus*	1	3.0 ± 0.4	2.5	Hexane	[49]	3.4	Ethyl acetate	[49]
*Cryptheconidium*	1	3.8 ± 4.2	0.9	Ethyl acetate	[48]	12.7	Hexane	[48]
*Desmodesmus*	1	7.8		Methanol	[8]			
*Dictyochloropsis*	1	38.5		Hexane, Chlorophorm, Ethyl Acetate, Ethanol 70%, Water	[55]			
*Dunaliella*	3	2.3 ± 2.1	0.1	Hexane	[62]	5.9	Methanol	[17]
*Fischerella*	1	2.3 ± 1.8	0.4	Hexane	[49]	5.0	Water	[49]
*Haematococcus*	1	0.9 ± 0.3	0.5	Ethanol/Water 3:1	[20]	1.2	Ethanol/Water 3:1	[20]
*Isochrysis*	2	2.3 ± 1.3	0.2	Water	[62]	3.9	Methanol	[18]
*Microchaete*	1	1.5 ± 1.5	0.2	Hexane	[49]	3.7	Water	[49]
*Nannochloropsis*	4	2.9 ± 3.1	0.1	Water	[62]	8.0	Acetone	[56]
*Neochloris*	1	9.8		Acetone	[56]			
*Nitzschia*	1	2.9 ± 0.7	2.4	Ethyl acetate or Hexane	[48]	3.9	Water	[48]
*Nostoc*	3	6.4 ± 8.9	0.3	Hexane	[49]	39.9	Hexane	[48]
*Oscillatoria*	2	17.4 ± 6.2	8.0	Water	[54]	24.5	Hexane, Chlorophorm, Ethyl Acetate, Ethanol 70%, Water	[55]
*Parachlorella*	1	1.4		Ethanol/Water 3:1	[20]			
*Pavlova*	1	0.2 ± 0.1	0.1	Hexane	[62]	0.3	Ethyl acetate	[62]
*Phaeodactylum*	3	4.2 ± 2.3	2.1	Ethanol/Water 3:1	[56]	9.9	Acetone	[56]
*Phormidium*	1	28.0		Hexane, Chlorophorm, Ethyl Acetate, Ethanol 70%, Water	[55]			
*Porphyridium*	1	6.5		Acetone	[56]			
*Rhodomonas*	1	0.0		Methanol or Hexane	[52]			
*Scenedesmus*	2	15.5 ± 13.6	1.9	Ethanol/Water 3:1	[20]	29.1	Hexane, Chlorophorm, Ethyl Acetate, Ethanol 70%, Water	[55]
*Schizochytrium*	1	3.3 ± 4.7	0.0	Ethyl acetate or Hexane	[48]	13.6	Hexane	[48]
*Skeletonema*	1	9.5 ± 2.9	6.6	Methanol	[57]	12.4	Methanol	[57]
*Synechococcus*	1	3.5 ± 1.5	2.1	Hexane	[48]	5.6	Ethyl acetate	[48]
*Tetraselmis*	4	2.8 ± 5.2	0.0	Hexane	[52]	20.0	Acetone	[56]
*Thraustochytrium*	1	2.2 ± 1.3	1.2	Ethyl acetate	[48]	4.0	Hexane	[48]
*Tolypothrix*	1	1.4 ± 1.2	0.2	Hexane	[49]	3.0	Water	[49]

#### 3.3.2. DPPH RSA

Three-way ANOVA showed significant effects of species, species–phase interaction, and species–solvent interaction on DPPH RSA. *C. minutissima* extracts at the early stationary phase presented significantly higher DPPH RSA than most other extracts, while methanolic extracts of *C. minutissima* showed significantly higher DPPH RSA than all other extracts. The highest and lowest DPPH RSA in aqueous extract was found in *D. salina* at the early stationary phase (1.32 ± 0.27 mg TE g^−1^ DW) and *N. oculata* at the early stationary phase (0.29 ± 0.29 mg TE g^−1^ DW). In methanolic extracts, *C. minutissima* showed the highest DPPH RSA at the early stationary phase (3.74 ± 0.23 mg TE g^−1^ DW, *p* < 0.05), while *D. salina* the lowest value at the late stationary phase (0.15 ± 0.15 mg TE g^−1^ DW).

#### 3.3.3. FRAP

Three-way ANOVA showed significant effects of species, species–solvent interaction, and phase–solvent interaction on FRAP. Methanolic extracts of *C. minutissima* showed significantly higher FRAP than all other extracts, with methanolic extracts of *N. oculata* following. Methanolic extracts at the early stationary phase showed significantly higher FRAP than all aqueous extracts. The highest and lowest FRAP in aqueous extracts was found in *T. lutea* at the late stationary phase (0.92 ± 0.06 mg g^−1^ DW) and *N. oculata* at the late stationary phase (0.21 ± 0.08 mg GAE g^−1^ DW), while in case of methanolic extracts *C. minutissima* showed the highest activity at the early stationary phase (18.24 ± 0.81 mg g^−1^ DW, *p* < 0.05) and *T. lutea* the lowest at the late stationary phase (1.47 ± 0.01 mg GAE g^−1^ DW).

#### 3.3.4. ICA

Three-way ANOVA showed only significant effect of species-solvent interaction on ICA. Methanolic extracts of *C. minutissima*, *N. oculata*, and *I. galbana* presented significantly higher ICA than aqueous extracts of *C. minutissima* and *D. salina*. The highest and lowest ICA in aqueous extract was found in *I. galbana* at the early stationary phase (16.74 ± 0.10 mg EDTA g^−1^ DW) and *C. minutissima* at the late stationary phase (8.84 ± 0.12 mg EDTA g^−1^ DW), while the highest and lowest ICA in methanolic extract was found in *C. minutissima* at the early stationary phase (19.61 ± 0.09 mg EDTA g^−1^ DW) and *T. lutea* at the late stationary phase (13.28 ± 0.57 mg EDTA g^−1^ DW).

#### 3.3.5. Pearson Correlation Analysis

DPPH RSA and ICA of aqueous extracts were negatively correlated, with a Pearson score −0.702 (Figure 4). ICA of aqueous and methanolic extracts was also negatively correlated with each other with a score of −0.821. In contrast, TPC of the methanolic extract had a high correlation with Chla and total Chl, and high correlation with the FRAP and DPPH RSA of the methanolic extract. Moreover, FRAP and DPPH RSA of the methanolic extract were correlated to Chla and carotenoids, as well as with each other with a lower score than with TPC.

#### 3.3.6. PCA

PCA showed similar results to the Pearson test, indicating a strong correlation between TPC and DPPH RSA of the methanolic extracts and a positive correlation with total chlorophyll (Figure 5). Moreover, it assigns high TPC and DPPH RSA of methanolic extracts to *C. minutissima* and high lipid content to *N. oculata* in agreement with the ANOVA results.

The lack of correlation between TPC and DPPH RSA of aqueous extracts found in this study is in agreement to [48], while there was a strong correlation in case of the methanolic extracts in agreement to others [8,62]. In Appendix A, TPC, FRAP, and DPPH RSA of the current study data are plotted against those of Li et al. and others. The strong correlation between Chl and phenolic content, and to a lesser extent to DPPH RSA of the methanolic extracts suggests that determination of phenolics through the Folin–Ciocalteu method should be carefully interpreted in microalgae. While authors have correlated concentration of specific phenolics to the antioxidant capacity [15], it has been argued that other compounds such as carotenoids, which are abundant in the cell, contribute to the observed capacities of the extracts [12,20]. However, Chl is usually not taken into consideration, despite being the most abundant pigment in microalgal cells and having been identified as a major source of interference in the F-C assay in plants [64]. Martins et al. has reported a correlation between Chl content and ABTS RSA but not with DPPH RSA [7].

### 3.4. Chromatographic Analysis

Initially, extracts of *C. minutissima*, *D. salina*, *N. oculata*, and *I. galbana* prepared with Protocol B (70% MeOH) were analyzed with HPLC and LC-MS as described in Section 2.8.1 and Section 2.8.2 in order to obtain their fingerprinting. The small amounts of available biomass did not allow for analytical characterization. All the HPLC chromatograms presented two distinct sets of peaks at 260 nm, one at the beginning representing compounds not retained by the C18 column, probably including aromatic aminoacids and peptides, and one set at the end (100% acetonitrile), where hydrophobic pigments were expected to elute (Appendix A). The second set of peaks was also visible at 405 nm (or 425 nm for *I. galbana*, Appendix A), which confirms the presence of chlorophylls and carotenoids. Only *D. salina* and *I. galbana* presented small peaks at 260 nm in the middle of the chromatograms which indicates the presence of phenolic compounds in very low concentration, but none of them was detected with LC-MS in the negative ionization mode. In the 70% hydromethanolic extract of *C. minutissima* some peaks were detected with the MS detector at around 40% acetonitrile which had *m*/*z* > 1000; those were absent when *C. minutissima* was extracted with 100% methanol (protocol C) and might be peptides (Appendix A). Regarding *N. oculata*, no peaks of phenolic nature were recorded; a peak probably originating from non-aromatic peptides and numerous peaks corresponding to pigments was present (Appendix A). In agreement with this observation, we have recently fully characterized its chemical composition showing the presence of 14 different carotenoids (the major ones were violaxanthin and antheraxanthin) and Chla [65]. Only *T. lutea* displayed peaks of phenolic nature in considerable amounts in both HPLC (260 nm) and LC-MS (Appendix A); those were evident even though the biomass was extracted with 100% methanol (protocol C). In an attempt to better characterize *T. lutea*, we ran the analysis on a semi-preparative HPLC C18 column and prolonged the LC-MS elution program (Appendix A), which improved separation of the observed peaks. With LC-MS and the modified elution program, we could separate and identify standard phenolic compounds commonly reported to be present in microalgae (Appendix A): gallic acid, p-coumaric acid, protocatechuic acid, 4-hydroxybenzoic acid, vanillic acid, (−)-epigallocatechin 3-gallate, (+)-catechin, caffeic acid, trans-cinnamic acid, hydroxybenzaldehyde, salicylic acid, (−)-epicatechin, (−)-epicatechin 3-gallate and resveratrol. However, none of them was present in the extract of *T. lutea*.

In summary, the main compounds in the methanolic extracts were pigments (chlorophylls and carotenoids), along with fatty acids, which were not measured in the extracts but are known to be dissolved in methanol [66]. The aqueous extracts might contain peptides, carbohydrates, very low amounts of phenolics in some cases, and other hydrophilic compounds. Those observations explain our finding that in the aqueous extracts (Protocol A) there is a lack of correlation between TPC and the antioxidant assays in contrast to the methanolic extracts (Protocol A), which displayed TPC and antioxidant activity proportional to the pigment concentration in the biomass. The above suggests that pigments in the methanolic extracts and other polar compounds in the aqueous extracts might be more relevant to the TPC registered with the F-C assay and the observed antioxidant activities than phenolic compounds. It is important to mention that an interference of pigments with the F-C assay would not relate to the absorption spectrum of the pigments, since the absorption of the reduced F-C reagent is measured at 760 nm. It is well-known that F-C assay is non-selective, determining a wide variety of compounds and that TPC is representative of the antioxidant activity in extracts lacking phenolics [67,68].

## 4. Conclusions

*C. minutssima*, *D. salina*, *N. oculata, I. galbana,* and *T. lutea* were cultivated in batch mode under saturating light intensity and continuous aeration. Under the studied conditions, *C. minutissima* and *N. oculata* displayed the highest biomass productivity. Composition analysis of the biomass obtained at the late stationary phase showed high concentration of protein in all species and greater lipid content of *N. oculata*, while *C. minutissima* contained the highest pigment content. The only species containing both DHA and EPA were *T. lutea* and *I. galbana,* whereas only *N. oculata* produced EPA in significant amounts. *D. salina* and *I. galbana* displayed the highest TPC in the aqueous extracts as well as some HPLC peaks that could indicate low presence of phenolics. That was not reflected in the antioxidant activity however, which did not correlate with TPC in the aqueous extracts. On the contrary, TPC and antioxidant activity of methanolic extracts correlated with pigments, especially chlorophyll, with their presence being evident both visually and chromatographically. Finally, *T. lutea* presented possible phenolic peaks in both HPLC and LC-MS, which could not be identified.

In conclusion, despite the possible presence of phenolics in some of the species, it is evident that pigments, proteins, and fatty acids should be the main target for optimization and thus *N. oculata* and *T. lutea* are recommended for further research, the first due to its high protein and EPA content and the second due to its protein, DHA, and carotenoid content. *N. oculata* however has the great advantage of robustness and productivity, which is especially important for large-scale production. Another concluding remark is that it is highly recommended to avoid making conclusions regarding the phenolic content of microalgae using the Folin–Ciocalteu assay, since it is non-selective and affected by peptides and pigments, both abundant in microalgae. Our finding that TPC and antioxidant activity of methanolic extracts are highly correlated to the presence of chlorophyll will hopefully help to prevent further confusion regarding the phenolic content of microalgae.

## Figures and Tables

**Figure 1 antioxidants-11-01320-f001:**
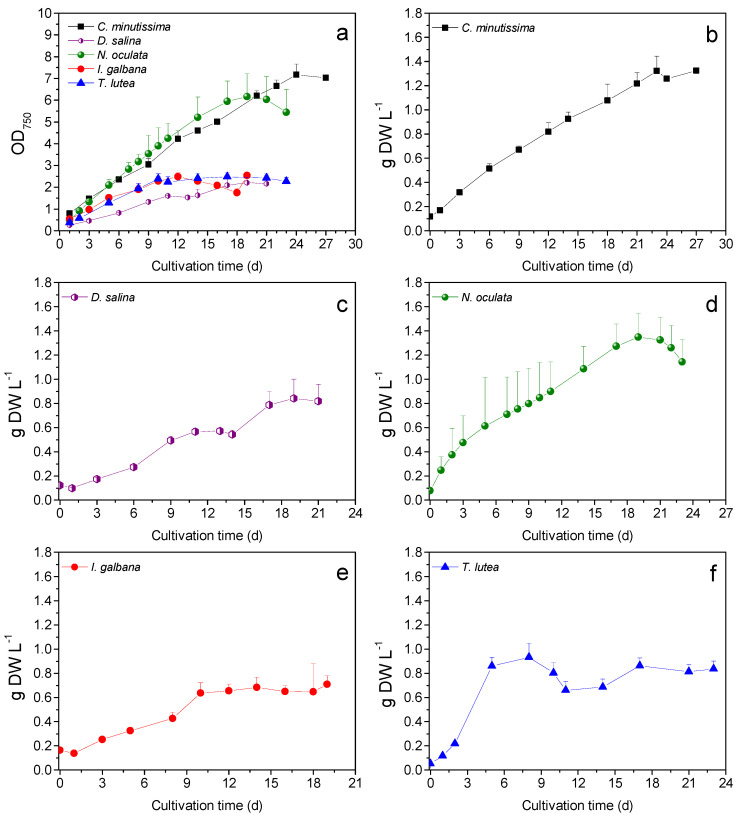
OD_750_ (**a**), and TSS concentration values upon cultivation of the marine microalgal species *C. minutissima* (**b**), *D. salina* (**c**), *N. oculata* (**d**), *I. galbana* (**e**), and *T. lutea* (**f**).

**Figure 2 antioxidants-11-01320-f002:**
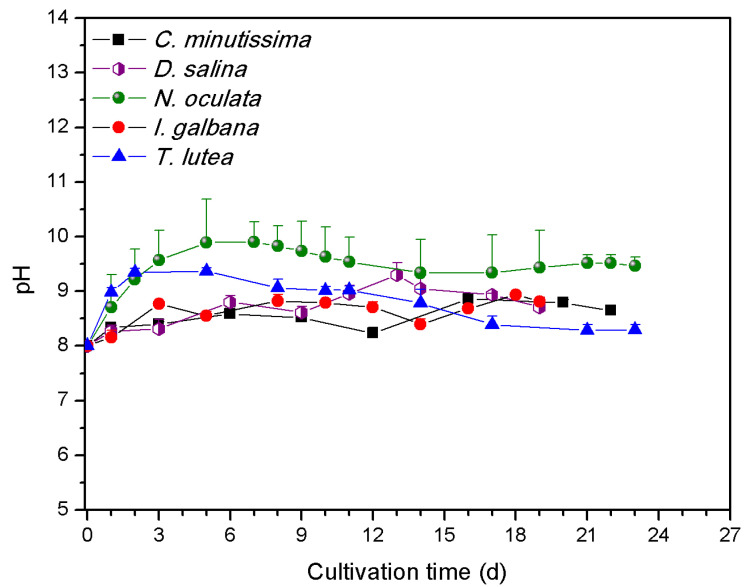
pH values upon cultivation of the marine microalgal species *C. minutissima*, *D. salina*, *N. oculata, I. galbana*, and *T. lutea*.

**Figure 3 antioxidants-11-01320-f003:**
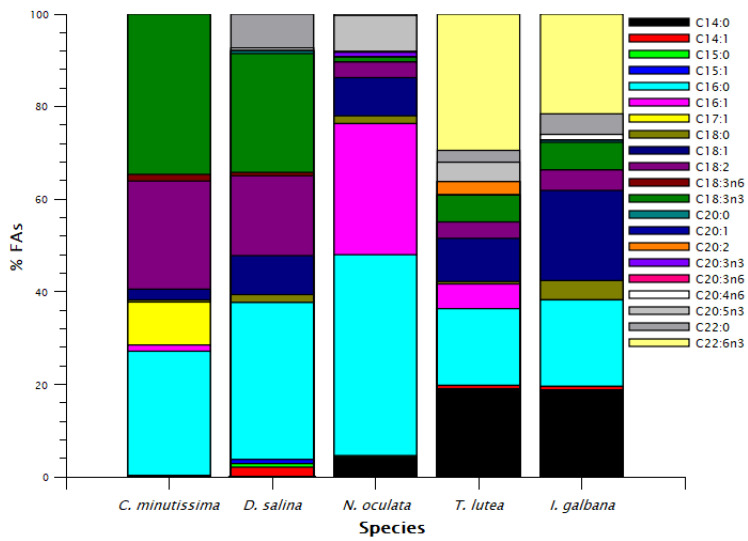
FA profile during the late stationary phase. Concentrations of FAs are expressed as percentage of total FAs.

**Figure 4 antioxidants-11-01320-f004:**
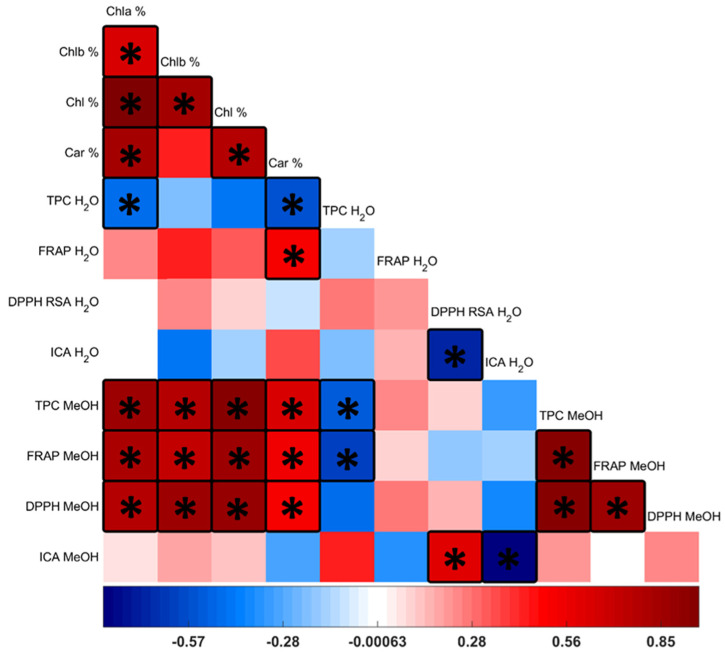
Correlation matrix (Pearson). Color bar at the bottom indicated the Pearson correlation coefficient value. Stars (*) indicate difference from 0 with a significance level alpha = 0.05. Figure created in Matlab with code provided by Layden et al. [63].

**Figure 5 antioxidants-11-01320-f005:**
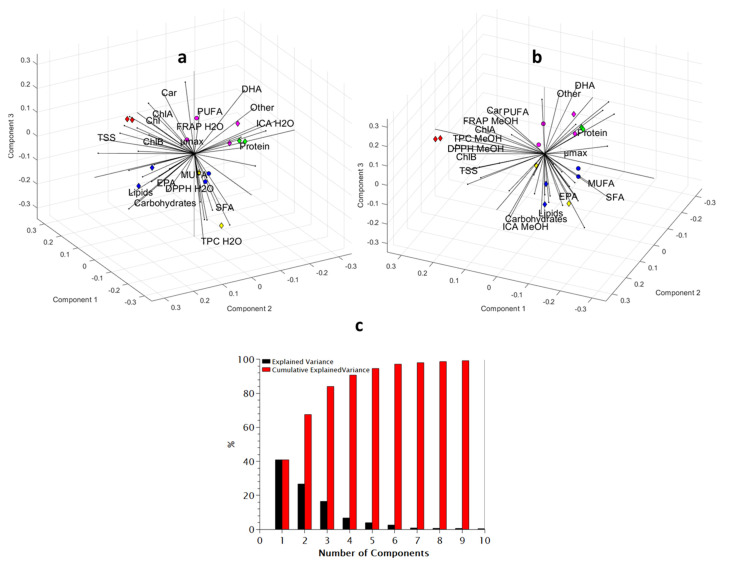
(**a**,**b**) *** **** Biplots of the first three PCA components on proximate composition (84.07% of the explained variance). EPA and DHA here represent the percentage in the dry biomass, while SFA, MUFA, and PUFA are percentages in the total lipids. Markers represent data (color indicates species and marker type indicates growth phase) and are grouped by similarity. The variation of data is expressed by the position of datapoints in a three-dimensional plot of three components that are composed of all the characteristics under analysis. Lines that originate from the origin of the graph represent the characteristics used to synthesize the principal components. The length of each line is proportional to the contribution of the corresponding characteristic to the principal components. The angle between two lines indicates positive (close to 0°), negative (close to 180°) or no (close to 90°) correlation between the variables. Red: *C. minutissima*, yellow: *D. salina*, green: *I. galbana*, blue: *N. oculata*, purple: *T. lutea***,** circles: early stationary phase**,** diamonds: late stationary phase. (**c**) The explained variance and explained cumulative variance as a function of the number of components. * TPC and antioxidant data of aqueous and methanolic extracts have been omitted from a and b, respectively, for clarity. ** Other data used for PCA were also omitted for clarity. The full data set and 3D figure can be found in the Appendix A.

**Table 1 antioxidants-11-01320-t001:** Nutrient consumption, pH, final OD_750_, final TSS, and μ_max_ ± the standard deviation.

Species *	*C. minutissima*	*D. salina*	*N. oculata*	*T. lutea*	*I. galbana*
**mg N L^−1^ conc.**	48.05 ± 0.10 ^ab^	44.91 ± 1.42 ^b^	49.44 ± 0.33 ^a^	49.77 ± 0.02 ^a^	48.96 ± 0.06 ^a^
**μg P L^−1^ conc.**	4605.95 ± 0.05 ^a^	4605.66 ± 0.01 ^b^	4605.97 ± 0.03 ^a^	4605.80 ± 0.02 ^ab^	4605.69 ± 0.04 ^b^
**Final pH**	8.65 ± 0.05 ^ab^	8.55 ± 0.25 ^ab^	9.50 ± 0.11 ^a^	8.31 ± 0.07 ^b^	8.62 ± 0.28 ^ab^
**OD_750_/Dilution**	6.85 ± 0.18 ^a^	2.17 ± 0.35 ^b^	6.21 ± 0.69 ^a^	2.24 ± 0.16 ^b^	2.52 ± 0.12 ^b^
**TSS g DW L^−1^**	1.35 ± 0.13 ^a^	0.82 ± 0.14 ^ab^	1.25 ± 0.05 ^a^	0.83 ± 0.02 ^ab^	0.69 ± 0.10 ^b^
**μ_max_ (d^−1^)**	0.31 ± 0.04 ^a^	0.28 ± 0.18 ^a^	0.45 ± 0.05 ^a^	0.45 ± 0.11 ^a^	0.29 ± 0.04 ^a^

* Absence of common lowercase letter superscripts in the same row indicates significant differences for each variable (Tukey–Kramer Test).

**Table 2 antioxidants-11-01320-t002:** Biomass composition, lipid analysis and EPA/DHA content of *C. minutissima*, *D. salina*, *N. oculata* and *T. lutea* during the late stationary phases. Values are given ± the standard deviation.

Species ***	*C. minutissima*	*D. salina*	*N. oculata*	*T. lutea*	*I. galbana*
**Ash (%)**	8.30 ± 1.54 ^de^	10.08 ± 0.01 ^cde^	3.81 ± 0.21 ^e^	12.81 ± 3.16 ^bcde^	10.85 ± 0.35 ^cde^
**Carbohydrates (%)**	18.00 ± 3.43 ^abcde^	20.32 ± 2.48 ^abcde^	21.48 ± 10.62 ^abcde^	10.04 ± 1.67 ^ce^	8.81 ± 2.59 ^de^
**Protein (%)**	31.08 ± 0.93 ^abcd^	36.72 ± 5.05 ^ab^	27.71 ± 3.12 ^abcde^	39.69 ± 5.03 ^ab^	31.40 ± 4.32 ^abcd^
**Lipids (%)**	11.11 ± 0.15 ^cde^	9.23 ± 3.24 ^de^	36.70 ± 3.30 ^ab^	7.39 ± 0.45 ^de^	8.77 ± 0.56 ^de^
**Other (%)**	26.63 ± 5.46 ^abcde^	22.23 ± 5.44 ^abcde^	9.91 ± 9.91 ^cde^	28.35 ± 2.95 ^abcde^	44.06 ± 11.65 ^a^
**Chla (%)**	2.80 ± 0.16 ^a^	0.88 ± 0.24 ^bc^	0.71 ± 0.31 ^bcd^	0.95 ± 0.06 ^bc^	1.01 ± 0.25 ^bc^
**Chlb (%)**	1.01 ± 0.05 ^bc^	0.29 ± 0.04 ^cd^	0.00 ± 0.00 ^d^	0.00 ± 0.00 ^d^	0.08 ± 0.03 ^d^
**Car (%)**	1.07 ± 0.08 ^b^	0.24 ± 0.08 ^cd^	0.35 ± 0.12 ^bcd^	0.77 ± 0.07 ^bcd^	0.27 ± 0.02 ^cd^
**MUFA (% lipids)**	3.86 ± 0.84 ^f^	11.42 ± 0.45 ^ef^	36.58 ± 0.35 ^bcd^	15.36 ± 3.10 ^def^	22.38 ± 2.16 ^cde^
**PUFA (% lipids)**	59.44 ± 0.57 ^a^	44.27 ± 6.63 ^abc^	13.45 ± 0.50 ^ef^	45.86 ± 6.25 ^ab^	29.40 ± 3.81 ^bcd^
**SFA (% lipids)**	27.42 ± 0.34 ^bcde^	44.31 ± 6.19 ^abc^	49.97 ± 0.86 ^ab^	38.79 ± 9.35 ^abc^	48.22 ± 1.64 ^ab^
**EPA (mg g^−1^ DW)**	0.00 ± 0.00 ^c^	0.00 ± 0.00 ^c^	28.19 ± 0.03 ^a^	2.91 ± 2.9 ^c^	1.97 ± 1.97 ^c^
**DHA (mg g^−1^ DW)**	0.00 ± 0.00 ^c^	0.00 ± 0.00 ^c^	0.00 ± 0.00 ^c^	21.56 ± 2.04 ^ab^	14.93 ± 2.76 ^b^

* Absence of common letter superscripts in each group indicate significant differences (Tukey–Kramer test). Groups: (1) Ash, Carbohydrates, Protein, Lipids, Other (2) Chla, Chlb, Car (3) MUFA, PUFA, SFA, (4) EPA, DHA.

**Table 3 antioxidants-11-01320-t003:** Biomass composition, lipid analysis and EPA/DHA content of *N. oculata* and *T. lutea* during the early stationary phase. Values are given ± the standard deviation.

Species * ** ***	*N. oculata*	*T. lutea*
**Ash**	2.10 ± 0.61 ^e^	14.93 ± 2.90 ^cd^
**Carbohydrates (%)**	14.92 ± 1.83 ^cd^	7.21 ± 0.10 ^d^
**Protein (%)**	39.88 ± 1.72 ^ab^	43.30 ± 1.33 ^a^
**Lipids (%)**	26.39 ± 3.00 ^abc^	9.45 ± 0.30 ^d^
**Other (%)**	14.21 ± 3.48 ^abcd^	22.20 ± 4.15 ^cd^
**Chla (%)**	1.86 ± 0.11 ^aA^	1.95 ± 0.11 ^aA^
**Chlb (%)**	0.00 ± 0.00 ^c^	0.02 ± 0.00 ^c^
**Car (%)**	0.64 ± 0.13 ^b^	0.92 ± 0.02 ^b^
**MUFA (% lipids)**	31.68 ± 0.25 ^ab^	17.10 ± 0.81 ^b^
**PUFA (% lipids)**	19.38 ± 3.29 ^b^	45.71 ± 3.09 ^a^
**SFA (% lipids)**	48.94 ± 3.54 ^a^	37.19 ± 2.28 ^a^
**EPA (mg g^−1^ DW)**	33.74 ± 9.98 ^a^	2.80 ± 1.81 ^b^
**DHA (mg g^−1^ DW)**	0.00 ± 0.00 ^b^	31.31 ± 2.92 ^a^

* The variable groups of Table 2 also apply in this table. ** Absence of common lower case superscripts indicate significant difference between the two species at the early stationary phase (Tukey–Kramer test). *** Capital letters indicate significant difference of the presented value with its corresponding value from the late stationary phase for the same species (Tukey–Kramer test).

**Table 4 antioxidants-11-01320-t004:** TPC, FRAP, DPPH RSA, and ICA of aqueous and methanolic extracts during the early and late stationary phase. Values are given ± the standard deviation.

Species *	Phase	Solvent	TPC	FRAP	DPPH RSA	ICA
** *C. minutissima* **	**Early st.**	**H_2_O**	3.00 ± 0.30 ^def^	0.47 ± 0.04 ^d^	1.07 ± 0.18 ^c^	9.47 ± 0.12 ^abc^
		**MeOH**	9.04 ± 0.68 ^a^	18.24 ± 0.8 ^a^	3.74 ± 0.23 ^a^	19.61 ± 0.09 ^ab^
	**Late st.**	**H_2_O**	2.81 ± 0.24 ^def^	0.84 ± 0.03 ^d^	0.83 ± 0.02 ^c^	8.84 ± 0.12 ^c^
		**MeOH**	6.23 ± 0.52 ^abcd^	11.16 ± 0.35 ^b^	2.50 ± 0.05 ^b^	17.42 ± 0.18 ^abc^
** *D. salina* **	**Early st.**	**H_2_O**	6.44 ± 0.93 ^abc^	0.25 ± 0.03 ^d^	1.32 ± 0.27 ^c^	9.53 ± 0.34 ^abc^
		**MeOH**	2.25 ± 1.20 ^def^	3.84 ± 1.69 ^cd^	0.16 ± 0.14 ^c^	18.51 ± 0.50 ^ab^
	**Late st.**	**H_2_O**	8.78 ± 1.49 ^a^	0.34 ± 0.07 ^d^	0.70 ± 0.07 ^c^	9.60 ± 0.36 ^bc^
		**MeOH**	1.30 ± 0.37 ^ef^	2.04 ± 0.92 ^d^	0.15 ± 0.15 ^c^	18.13 ± 0.50 ^ab^
** *N. oculata* **	**Early st.**	**H_2_O**	3.46 ± 1.24 ^cdef^	0.29 ± 0.12 ^d^	0.29 ± 0.29 ^c^	13.58 ± 4.18 ^abc^
		**MeOH**	3.32 ± 0.10 ^cdef^	6.73 ± 0.73 ^c^	0.75 ± 0.13 ^c^	15.41 ± 3.53 ^abc^
	**Late st.**	**H_2_O**	3.81 ± 0.90 ^cdef^	0.21 ± 0.08 ^d^	0.51 ± 0.51 ^c^	13.69 ± 4.03 ^abc^
		**MeOH**	1.30 ± 0.26 ^ef^	3.63 ± 2.16 ^cd^	0.24 ± 0.04 ^c^	14.95 ± 2.63 ^abc^
** *T. lutea* **	**Early st.**	**H_2_O**	4.43 ± 0.57 ^bcdef^	0.55 ± 0.15 ^d^	0.90 ± 0.29 ^c^	14.34 ± 1.64 ^abc^
		**MeOH**	2.32 ± 0.36 ^def^	2.59 ± 0.43 ^cd^	0.36 ± 0.21 ^c^	16.10 ± 0.62 ^abc^
	**Late st.**	**H_2_O**	4.75 ± 0.53 ^bcde^	0.92 ± 0.06 ^d^	0.54 ± 0.08 ^c^	15.95 ± 0.48 ^abc^
		**MeOH**	1.25 ± 0.44 ^ef^	1.47 ± 0.01 ^cd^	0.25 ± 0.11 ^c^	13.28 ± 0.57 ^abc^
** *I. galbana* **	**Early st.**	**H_2_O**	8.13 ± 0.39 ^ab^	0.39 ± 0.50 ^d^	0.42 ± 0.04 ^c^	16.74 ± 0.10 ^abc^
		**MeOH**	2.03 ± 0.18 ^def^	3.05 ± 0.01 ^cd^	0.25 ± 0.16 ^c^	17.23 ± 0.15 ^ab^
	**Late st.**	**H_2_O**	6.49 ± 0.51 ^abc^	0.51 ± 0.27 ^d^	0.98 ± 0.19 ^c^	8.93 ± 0.76 ^abc^
		**MeOH**	1.78 ± 0.0 ^f^	1.58 ± 0.33 ^d^	0.62 ± 0.39 ^c^	19.04 ± 0.84 ^abc^

* In each column, average values without common letter superscripts differ significantly (Tukey–Kramer test).

## Data Availability

The data presented in this study are available in the article and Appendix A.

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
