# Peer review of "Total Phenolic Content, Biomass Composition, and Antioxidant Activity of Selected Marine Microalgal Species with Potential as Aquaculture Feed"

_antioxidants, 2022, doi:10.3390/antiox11071320_

Round 1

Reviewer 1 Report

The manuscript titled "Total Phenolic Content, Biomass Composition, And Antioxidant Activity Of Selected Marine Microalgal Species With Potential As Aquaculture Feed," and authored by Andriopoulos and colleagues, deals with the investigation of phenolic content and antioxidant properties of different species of algae grown in aquaculture. The manuscript is verily well-written, and covers a topic that can help round out the current state of the art. The weakness of this article, however, is the lack of chemical profiling through chromatographic methodologies to identify which compounds actually contributes to the obtain values observed by the authors. Moreover:

·         ABSTRACT SECTION: In my opinion, the abstract needs to be rewritten almost completely. As clearly stated in the guidelines for authors, this section should be a maximum of 200 words and should contain a brief description of the current state of the art, emphasizing the problem, the authors' main objective, a small description of the used methodologies, the obtained results, and a conclusion sentence. At this time, the abstract presented by the authors describes far too much about the used methodologies and the obtained results, without addressing the other points. Please fix it.

·         KEYWORDS SECTION: regarding the keywords, they are a useful tool to help indexers and search engines to find relevant papers of interest. If scientific search engines (such as PubMed, Scopus, Google Scholar, etc) can find a potential manuscript by the use of words contained in both title, abstract, and keywords. Consequently, readers will be able to find it too thank this words. An easier search of the manuscript allows to increase the number of people reading your manuscript after publication and, then, to obtain more citations. Consequently, keywords should be words preferably not contained in the title or abstract. This short explanation is to suggest that authors introduce as many keywords as they can, and replace those words that are already present at least in the title with new keywords properly related to the reviewed manuscript.

·         INTRODUCTION SECTION: The authors should better specify that the use of algae, or their extracts, is a very important topic in recent years. This is mainly due to both the use of these organisms for the preparation of both dietary supplements and biostimulant formulations in sustainable agriculture. In the first case, it would seem that what determines the beneficial effect derived from algae intake is related to the particular composition of the lipid fraction, including fatty acids (10.1016/j.jff.2019.103508; 10.3390/biom9110708; 10.1002/9781119542650.ch13). In particular, Scientific evidence has proven how algae administration in in vitro models is adept at modulating gene expression and enzymatic activity of various biological pathways, including ARE/Nrf2 pathway. In the second case, the beneficial effect would be mainly related to the presence of bioactive compounds originating from secondary metabolism, including many phenolic compounds, which would be able to imitate the action of plant hormones (10.1016/j.biotechadv.2021.107754; doi.org/10.3390/plants10030531; 10.3390/agriculture11060557).

·         MATERIAL AND METHODS SECTION: (i) section 2.3. need a reference; (ii) section 2.4. should be divided into additional subsections (2.4.1.; 2.4.2.; etc.) each reporting the extraction and analysis method comprehensively for each evaluated parameter

·         RESULTS AND DISCUSSION SECTION: (i) Tables in this section should be set up according to the style of the journal. In particular, standard deviations should not be placed in a separate column, but combined with the median value (i.e. 48.05+0.10ab); (ii) I do not understand why the standard deviations are exclusively reported for three bars. If it's a percentage composition, maybe they shouldn't be reported at all; (iii) Table 3 should be placed in the previous paragraph; (iv) Table 6 could be transformed into a figure, depicting the Pearson correlations as a square, assigning different colors (blue to red for example) for different correlation values.; (v)  at the bottom of the tables, the authors correctly reported footnotes. However, at least for those pertaining to statistical analysis, the authors should specify for which statistical test the significance was observed.

·        CONCLUSION SECTION: The authors should extend this section by better reporting the results observed during their experiments.

Reviewer 2 Report

The subject of the work is interesting and has a practical aspect, especially at a time when microalgae are becoming more and more popular in the world. However, the research aimed at assessing the chemical composition and antioxidant properties should be very accurate and I believe that the content of individual polyphenols should be tested, not just total polyphenols. I consider the use of analytical methods assessing only the content of a group of compounds in the tested material to be the weakness of the manuscript.

In the introduction to the manuscript, I propose to supplement the information on the polyphenolic composition and antioxidant activity of microalgae. Were the content of polyphenols given in previous publications, were individual compounds identified and attention was drawn to the fact that microalgae are their important source? I recommend that you provide a short information on this in the introduction to the article.

The analytical methods used are of different accuracy. The spectrophotometric methods used to assess the content of carotenoids, chlorophylls and polyphenols are less accurate and do not assess the content of individual compounds, which is a weak point of the manuscript. As the analytical methods for assessing the biomass composition are standardized, no more detailed description is needed.

In the description of the analytical methods, I recommend changing the description of the centrifugation conditions rmp to g, which is more precise and does not require information about the centrifuge.

There is a problem with the charts that needs to be corrected. The caption for figure 1 is before table 2. However, the graph 2 showing the changes in pH is not included.

Why in the table number 2 in all values the belonging to a given homogeneous group is not given (eg MUFA, for N. oculata, EPA for T.lutea and I. galbana)? It is clear that when the results do not differ statistically then no determinations are made, but why are the determinations missing for the results where significant differences were detected?

The marking of statistical differences in Table 3 seems non-standard, so I propose to describe in detail how the means were compared in the section describing the statistical methods. Different capital letters show the difference between the results in Tables 2 and 3? What are the differences in lower case letters? Between the tested parameters N. oculata and T.lutea? If so, why aren't they prescribed for T.lutea? This is a non-standard designation, therefore an explanation is required.

Why was the third factor, the development phase, not included in the markings in Table 4?

The compilation of information contained in previous publications on TPC content in microalgae and cyanobacteria is unusual for manuscripts presenting studies, not reviews (Table 5). However, in my opinion, this table should remain in the article, as it illustrates very well the differences in the TPC content obtained for different species.

Round 2

Reviewer 2 Report

All my comments have been included in the revised manuscript.